# Influence of Calcined Clay Pozzolan and Aggregate Size on the Mechanical and Durability Properties of Pervious Concrete

Kwabena Boakye  and Morteza Khorami *

School of Energy, Construction & Environment, Faculty of Engineering & Computing, Coventry University, Coventry CV1 5FB, UK; boakyek4@uni.coventry.ac.uk
* Correspondence: morteza.khorami@coventry.ac.uk

**Abstract:** Pervious concrete has been reported as a viable solution to reduce stormwater run-off, the heat-island effect, road noise, and pavement flooding. Previous researchers have focused on analysing the structural properties and functionality of pervious concrete. However, relatively few studies have been conducted into the addition of supplementary cementitious materials (SCMs), such as calcined clay, in pervious concrete and its effect on long-term durability. This paper has studied the effect of calcined clay pozzolan as a partial substitute for Portland cement in pervious concrete, together with the influence of coarse aggregate size. A water–binder ratio of 0.4 and aggregate–binder ratio of 4.0, as well as a superplasticiser content of 0.95%, were maintained for all mixes. Two sizes of coarse aggregates were used for this study: 9.5 mm and 20 mm. CEM-I cement was partly substituted with calcined clay in dosages of 0 to 30% in replacement intervals of 5%. The mechanical tests conducted included the split tensile test, compressive strength test, and flexural strength test. Durability measurements such as the rapid chloride permeability test (RCPT), thermal conductivity and sulphate resistance tests were also carried out. The mechanical properties of the pervious concrete followed a similar trend. The results showed that at 20% replacement with calcined clay, the compressive strength increased by 12.7% and 16% for 9.5 mm and 20 mm aggregates, respectively. The flexural strength improved by 13.5% and 11.5%, whereas the splitting tensile strength increased by 35.4% and 35.7%, respectively, as compared to the reference concrete. Beyond 20% replacement, the tested strengths declined. The optimum calcined clay replacement was found to be 20% by weight. Generally, pervious concrete prepared with 9.5 mm obtained improved mechanical and durability properties, as compared to those of 20 mm aggregates.

**Keywords:** calcined clay; pervious concrete; durability; aggregate size; thermal conductivity; flexural strength

## 1. Introduction

The use of pervious concrete is one of the solutions that have been proposed to curb some of the challenges associated with urban growth such as the heat island effect and flooding. Pervious concrete is a specially designed kind of concrete with significant porosity to enable water and other liquids to freely drain from the surface of slabs [1]. One major characteristic of pervious concrete is the presence of numerous interconnecting voids created due to little or no sand in its design and production, contrary to normal concrete [2]. This property allows pervious concrete to provide a range of environmental advantages, including lowering runoff, raising groundwater levels, minimising the impact of urban heat islands, wastewater treatment, skid resistance improvement, and traffic noise absorption [3,4]. The distinctive porous structure and low mechanical property of pervious concrete allow it to be used for construction application in areas such as basketball courts, tennis courts, open car parking areas, pedestrian walkways, and pavements.

Supplementary cementitious materials (SCMs), over the years, have been introduced into pervious concrete systems to influence their properties and performance. SCMs such

as flay ash, calcined clay, metakaolin, silica fume, and ground granulated blast furnace slag (GGBS), apart from the environmental and energy saving benefits [5], are known to improve workability, reduce porosity, control bleeding, prevent segregation, increase long-term strength, and generally improve the durability of concrete [6–9]. During pozzolanic reactions, silicates in SCMs react with $Ca(OH)_2$ emanating from the hydration of cement to produce further cementitious compounds in the form of calcium silicate hydrates (C-S-H), needed for strength development [10].

A few researchers have investigated the effect of SCMs on the properties of pervious concrete and have reported varying results based on additional parameters such as mix design, methodology, and materials. Adil et al. [11] studied the properties of pervious concrete containing silica fume and reported an improved rheological, mechanical, and durability behaviour with as low as 5.5% silica fume replacement. Additionally, samples containing silica fume were found to demonstrate superior bonding characteristics with the binder paste than the reference samples. The optimum metakaolin was found to be 15%, based on the results obtained for strength determination. Researching the role of metakaolin in pervious concrete systems, Singh and Murugan [12] observed an increase in density, reduction in porosity, and increase in flexural as well as compressive strength, as the metakaolin content increased in the mix. Saboo et al. [13] experimented on the use of two different SCMs (fly ash and metakaolin) in pervious concrete. Portland cement was replaced by 5%, 10%, 15%, and 20% with fly ash and was also replaced with 2% metakaolin. The results indicated that the incorporation of 2% metakaolin caused a significant improvement in density, whereas porosity was reduced. A similar trend was observed in samples containing fly ash. The inclusion of SCMs was highly recommended to improve strength while maintaining cement content.

One important factor that influences the properties of pervious concrete aggregate size is the aggregate-to-binder ratio [14]. This could adversely impact the porosity as well as permeability and consequently affect strength and durability characteristics of pervious concrete. It has been observed by several researchers [15–21] that two distinct properties that influence the performance of pervious concrete are compressive strength and porosity. These properties can, however, be greatly affected by the size and type of coarse aggregate used in the production of the light-weight aggregate and pervious concrete [22].

The durability of concrete is its capacity to withstand abrasion, weathering action, and chemical attack while retaining the necessary technical qualities [23]. Depending on the exposure environment and intended qualities, different concretes require varying levels of durability [7]. As pervious concrete, by its design, is a porous material, practical steps should be considered to ensure its long-term durability. To examine the durability performance of pervious concrete, Taheri et al. [24] observed that the size of coarse aggregate could not significantly affect the mass of the pervious concrete after freeze and thaw cycles. When the cement content in pervious concrete was replaced with 5–20% rice husk ash (RHA) and calcium carbide waste (CCW), Adamu et al. [25] observed that these supplementary cementitious materials negatively affected the durability. The researchers, however, recommended that 0% RHA and 5% CCW could be utilised to achieve the optimum durability performance. According to Bilal et al. [26]], it may be possible to improve the mechanical and durability performance of pervious concrete with acceptable permeability and porosity by replacing silica fume, metakaolin, and binary combinations of silica fume and metakaolin from 5 to 10%. Based on an improvement in the mechanical and freeze–thaw durability performance of pervious concrete with appropriate functional qualities, the optimum SCM percentage replacement was discovered to be 10%.

Many laboratory and field investigations [1,2,11–13,24–28] have been conducted in the past on the functionality of pervious concrete mixtures. Researchers have previously studied the effects of altering variables such as the water–binder ratio, aggregate type, and their influence on properties such as density, porosity, and strength. A limited number of studies have also looked at how pervious concrete mixtures are affected by SCMs such as silica fume [11,26], metakaolin [12], fly ash [13], and incineration bottom ash (IBA) [29].

One SCM that has recently been promoted for construction application is calcined clay because of the availability of clay in almost every region in the world. However, little study has been conducted on the use of calcined clay in pervious concrete, focusing on its influence on mechanical and durability properties. Again, whether these properties of pervious concrete incorporated with calcined clay could also be affected by aggregate gradation remains to be investigated.

## 2. Materials and Methods

### 2.1. Materials

CEM I cement of Class 52.5 N, complying with BS EN 197-1 requirements, was used in this study. The mineralogical composition of the cement was found to be 60.3% $C_3S$, 14.2% $C_2S$, 6.8% $C_3A$, and 8.6% $C_4AF$. The calcined clay was obtained by heat-treating an impure kaolinitic clay at a temperature of 800 °C for 2 h, adopting a heating rate of 20 °C/h to ensure complete dihydroxylation of the kaolinite mineral. Table 1 presents the chemical compositions of the cement and calcined clay using X-ray fluorescence (XRF). The X-ray diffraction (XRD) spectra showing the mineralogical make-up of the cement and calcined clay are also shown in Figure 1.

**Table 1.** Chemical compositions of calcined clay and cement.

| Oxides, % | SiO$_2$ | Al$_2$O$_3$ | Fe$_2$O$_3$ | MgO | CaO | Na$_2$O | K$_2$O | MnO | TiO$_2$ | P$_2$O$_5$ | Cl | SO$_3$ | LOI |
|---|---|---|---|---|---|---|---|---|---|---|---|---|---|
| Calcined clay | 62.77 | 18.71 | 11.68 | 1.89 | 0.25 | 0.03 | 2.12 | 0.46 | 0.41 | 0.03 | – | 0.19 | 1.46 |
| Cement | 18.88 | 3.57 | 3.36 | 1.89 | 59.64 | 4.7 | 2.12 | 0.14 | 0.14 | 0.22 | 0.01 | 4.93 | 0.4 |

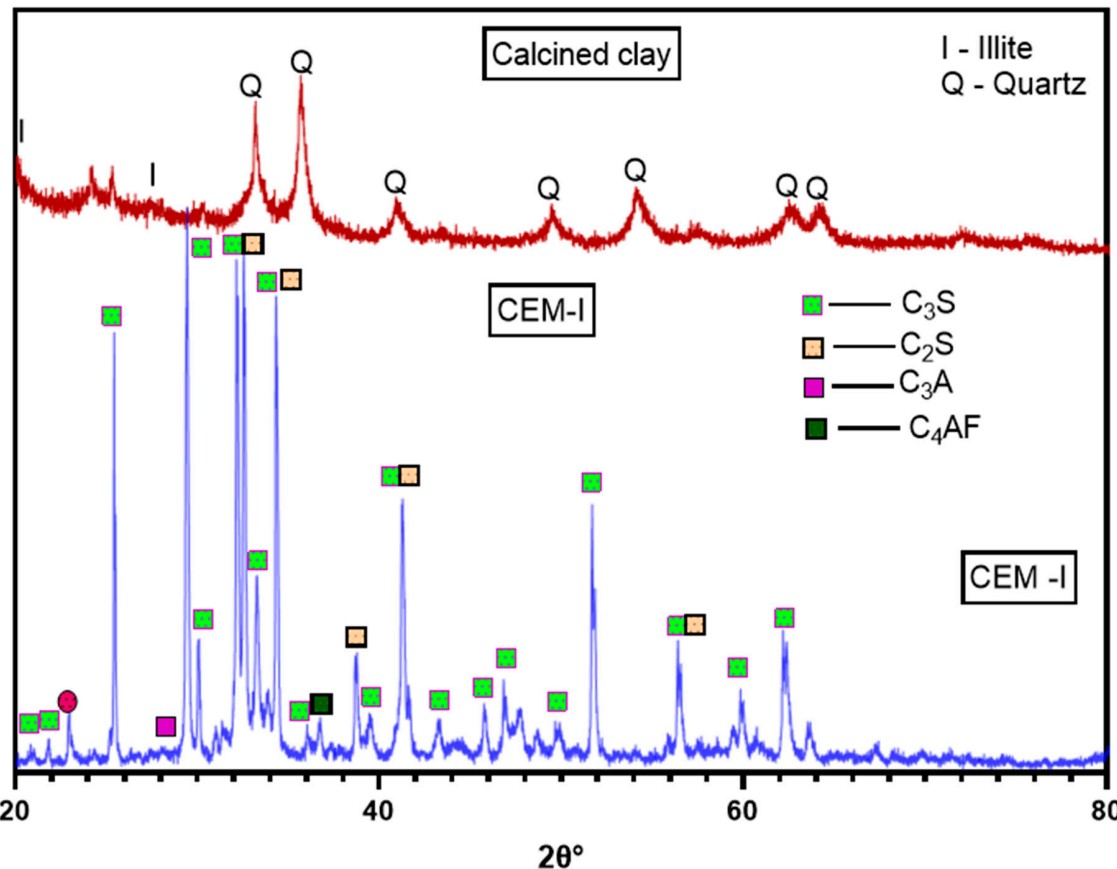

**Figure 1.** XRD of cement and calcined clay.

Two nominal sizes of coarse aggregates were used for this study: 9.5 mm and 20 mm. The grading curves of the coarse aggregates are presented in Figure 2. Other properties of

the coarse aggregates are shown in Table 2. Portable water having a pH of 7.0 was used in the preparation of the concrete as well as curing. Auracast 400, manufactured by FOSROC and having a solid content of 35 ± 2%, was the superplasticiser used in achieving the workability of the mixes.

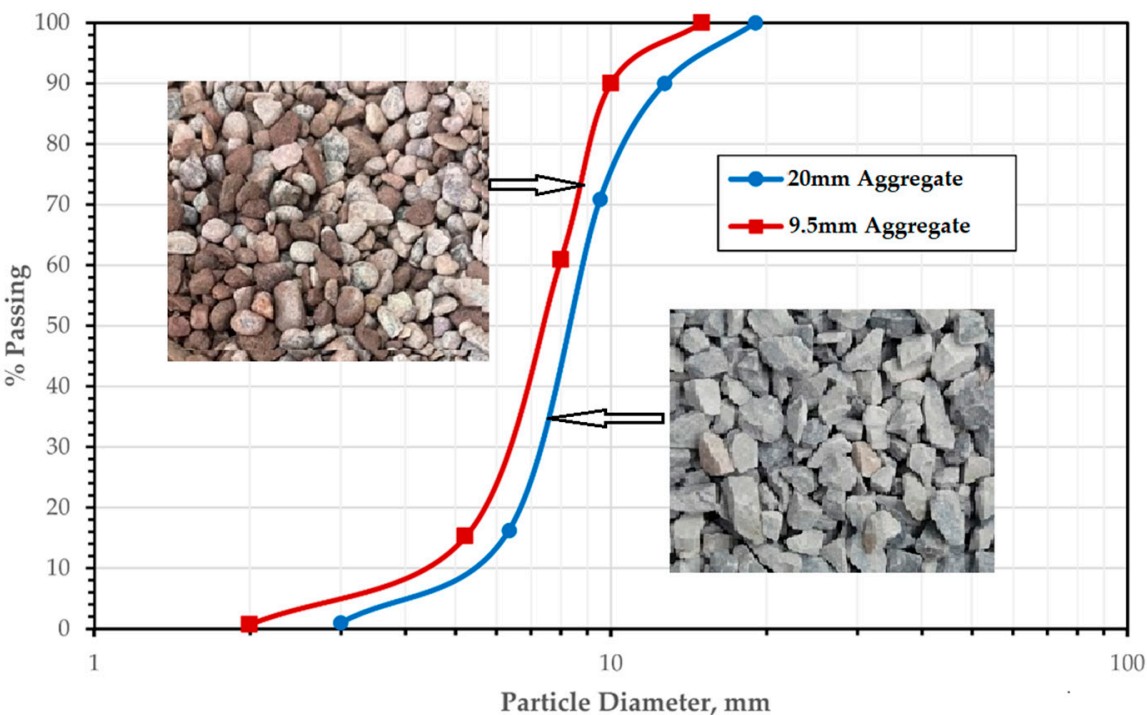

**Figure 2.** Particle size distribution of coarse aggregates.

**Table 2.** Physical properties of coarse aggregates.

| Aggregate Size | Specific Gravity | Crushing Value | Flakiness Index | Elongation Index | Impact Value | Combined Index |
|---|---|---|---|---|---|---|
| 9.5 mm | 2.65 | 23.4 | 17.0 | 10.0 | 11.0 | 13.05 |
| 20 mm | 2.41 | 25.3 | 16.5 | 8.6 | 11.5 | 12.8 |

The mix ratio of the concrete was designed in accordance with ACI 522R-10 [30]. To investigate the influence of these cementitious materials on pervious concrete, a water–binder ratio of 0.4 and aggregate–binder ratio of 4.0 were maintained, as well as a superplasticiser content of 0.95% (obtained by trial and error). Table 3 presents the details of the mix design.

**Table 3.** Mix proportion of pervious concrete.

| Sample ID | Cement | Calcined Clay | 9.5/20 mm Aggregate | Water | Superplasticiser |
|---|---|---|---|---|---|
| | (kg) | (kg) | (kg) | (kg) | (kg) |
| 0% (control concrete) | 1.5 | – | 5.4 | 0.525 | 0.075 |
| 5% | 1.425 | 0.075 | 5.4 | 0.525 | 0.075 |
| 10% | 1.35 | 0.15 | 5.4 | 0.525 | 0.075 |
| 15% | 1.275 | 0.225 | 5.4 | 0.525 | 0.075 |
| 20% | 1.2 | 0.3 | 5.4 | 0.525 | 0.075 |
| 25% | 1.125 | 0.375 | 5.4 | 0.525 | 0.075 |
| 30% | 1.05 | 0.45 | 5.4 | 0.525 | 0.075 |

*2.2. Methods*

Sample Preparation

Per the ACI 522R-10 standards, the raw materials were batched and mixed in a motorised concrete mixer for 3 min. The concrete was placed in 100 mm $\times$ 100 mm $\times$ 100 mm steel moulds in three layers and tamped 25 times and afterwards vibrated on a vibrating table to attain even distribution and compaction. The concrete was left on a shelf, demoulded after 24 h, cured under water (of temperature 23 °C), and crushed after 28 days for compressive strength determination. The compressive strength for each mix was tested in triplicate and the average strength recorded. Beams of size 500 $\times$ 150 $\times$ 150 mm were prepared to determine the flexural strength with reference to ASTM C78 [31]. Cylinders of diameter 100 mm and length 200 mm were moulded for the splitting tensile test per ASTM C497 [32].

The slump test, which is frequently used on normal concrete, does not accurately describe the workability of pervious concrete [12]. The rheology test method was, therefore, used to measure the workability of the pervious concrete mixes using a rotational rheometer. The vane spindle had an initial and final speed of 0.5 and 0.05 rev/s, respectively. The yield stress was then determined using the greatest torque recorded during the test.

The densities of the pervious concrete cube samples were determined using a specific gravity balance. In this test, the weight of each specimen was measured, both in air and water. The density of the porous concrete was calculated using Equation (1) where $V$ is the volume of concrete, P is density, m is mass, and pc is pervious concrete.

$$P_{pc} = \frac{m_{pc}}{V} = \frac{m_{pc}}{m_{pc} - m_{water}} \tag{1}$$

The porosity, on the other hand, was calculated using Equation (2) [33], where $V_T$ is the total volume of the specimen and $V_M$ is the change in volume of the sample because of voids.

$$n = \frac{(V_T - V_C)}{V_T} \tag{2}$$

Water permeability is an important property due to its relation to strength and the long-term durability of concrete. The permeability test in this investigation was carried out using the falling head permeameter as prescribed by ASTM C1701/M [34]. Permeability was calculated using Equation (3), where 'a' is the area of the standpipe, L is the height of the specimen, A is the area of the specimen, and t is the time between $h_1$ (initial water head) and $h_2$ (final water head). The setup of the permeability test is shown in Figure 3.

$$Permeability = \frac{aL}{At} \ln\left(\frac{h_1}{h_2}\right) \tag{3}$$

The pervious concrete cubes underwent a UPV test, which revealed information about the dense structure of the concrete specimen. A dense specimen with few pores has a high UPV value; the opposite is true for a sample with many pores. The ultraviolet pulse velocity test was conducted by estimating how long it takes for a pulse to go through the opposite sides of the pervious concrete cubes (approximately 100 mm) using the UPV equipment. The ultrasonic transducers were programmed to frequencies of 1 kHz. The velocity was calculated using Equation (4).

$$Velocity \ (km/s) = Distance \ travelled \ by \ the \ pulse \ time \tag{4}$$

With reference to the ASTM C 1202 RCPT procedure, electrical current was made to flow through sliced pervious concrete samples for a period of 6 h. Two slices of the same sample were placed in sodium chloride and sodium hydroxide solutions. At the ends of the samples, a potential difference of 60 V was preserved. The ASTM C 1113-90 method was used to measure the thermal conductivity of the pervious concrete. For the sulphate

resistance test, samples were cured for 28 days in water and then immersed in 5%-Na$_2$SO$_4$ solutions. It was allowed to stand for 90 days, and the effect of the 5%-Na$_2$SO$_4$ solution on their respective compressive strength and weight loss was determined.

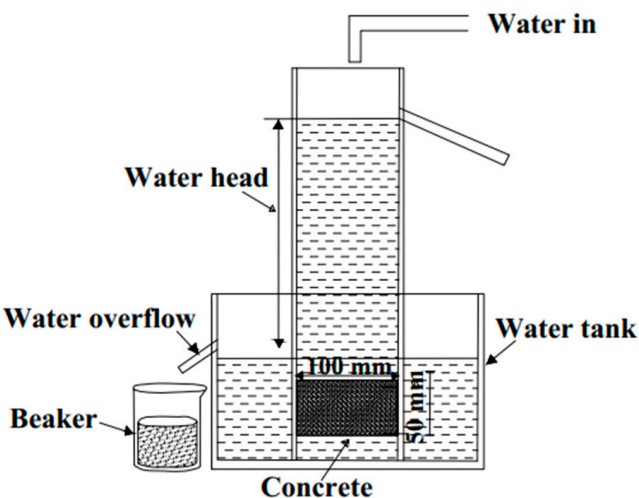

**Figure 3.** Setup for permeability test.

## 3. Results and Discussion

### 3.1. Rheology

The results showing the rheological properties of the calcined clay blended cement mixtures, according to the mix design, are presented in Figure 4. The reference concrete obtained a yield stress of 40 Pa, which increased by 38.5% when 5% calcined clay was added to the plain concrete. There was a further increase in yield stress until it reached its maximum at 15% calcined clay replacement. The viscosity, on the other hand, increased with increasing calcined clay replacement up to 20% replacement. There was no visible change in viscosity between 20% and 25%. A further increase in calcined clay content, however, caused the viscosity to increase.

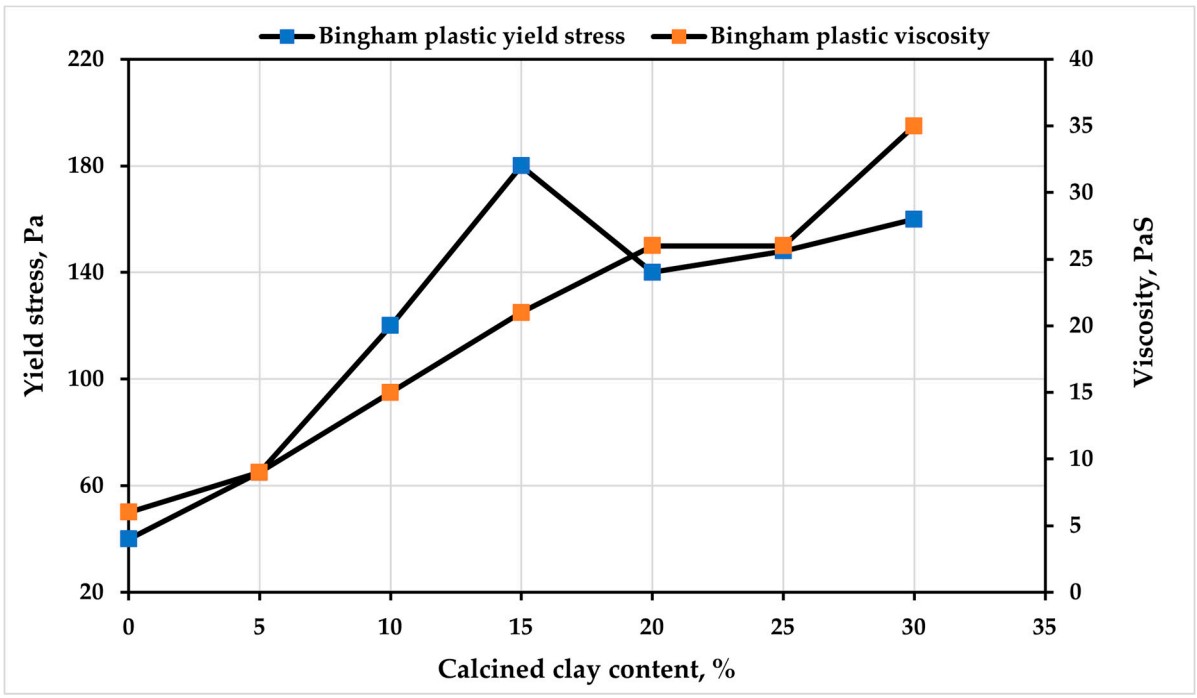

**Figure 4.** Rheology of paste containing varying calcined clay content.

*3.2. Density, Porosity, and Permeability*

The mechanical properties of Portland cement concrete are significantly affected by density and porosity, which ultimately impacts the ability of concrete to withstand harsh conditions and longevity [13]. The density of pervious concrete is greatly affected by aggregate size and shape. From Figure 5, as the aggregate size changed from 9.5 mm to 20 mm, the density was observed to decrease across all replacement levels. It was also observed that the density increased considerably with increasing calcined clay content for both aggregate types. At a maximum of 30% replacement, the density of the pervious concrete increased by 3.2% and 6.7% for 9.5 mm and 20 mm aggregates, respectively, after 28 days of curing.

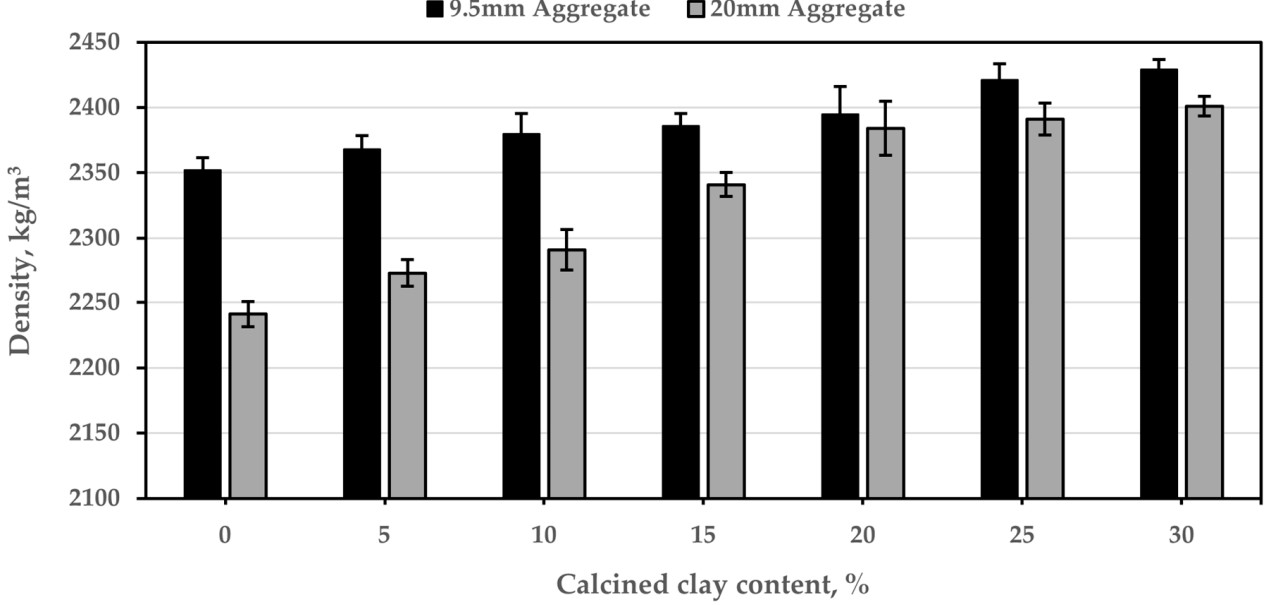

**Figure 5.** Density of pervious concrete.

As observed in Figure 6, the porosity decreased with the increase in calcined clay content for the mixes containing 9.5 mm and 20 mm aggregates, respectively. This could be attributed to the fine particles of the calcined clay, which are likely to occupy some of the pores in the concrete mix [35,36]. For both aggregate sizes, the porosity was highest in the control sample and lowest in the pervious concrete containing 30% calcined clay. Aggregate size was also observed to influence the degree of porosity in the concrete. The pervious concretes prepared with 9.5 mm aggregate was found to be less affected by porosity than 20 mm aggregate concretes, irrespective of the calcined clay content. On average, the porosity in mixes prepared with 20 mm aggregates was about 30.9% higher than that of 9.5 mm aggregates. This observation is consistent with results obtained by Saboo et al. [13], Singh and Murugan [12], and Shen et al. [29]. This is because the larger the aggregate size, the less it bonds with the mortar, thereby creating pores at the surface of contact [37–39]. The porosity in all batches was, however, found with the limits set by ACI 522R-10.

Similarly, as expected, permeability decreased as the dosage of calcined clay increased (shown in Figure 7). For mixes containing 30% calcined clay, permeability values were found to be about 39.2% and 31.5% (for 9.5 mm and 20 mm aggregates), respectively, less than those of the reference concrete samples.

This means that permeability was higher in samples containing 20 mm aggregates than 9.5 mm aggregates.

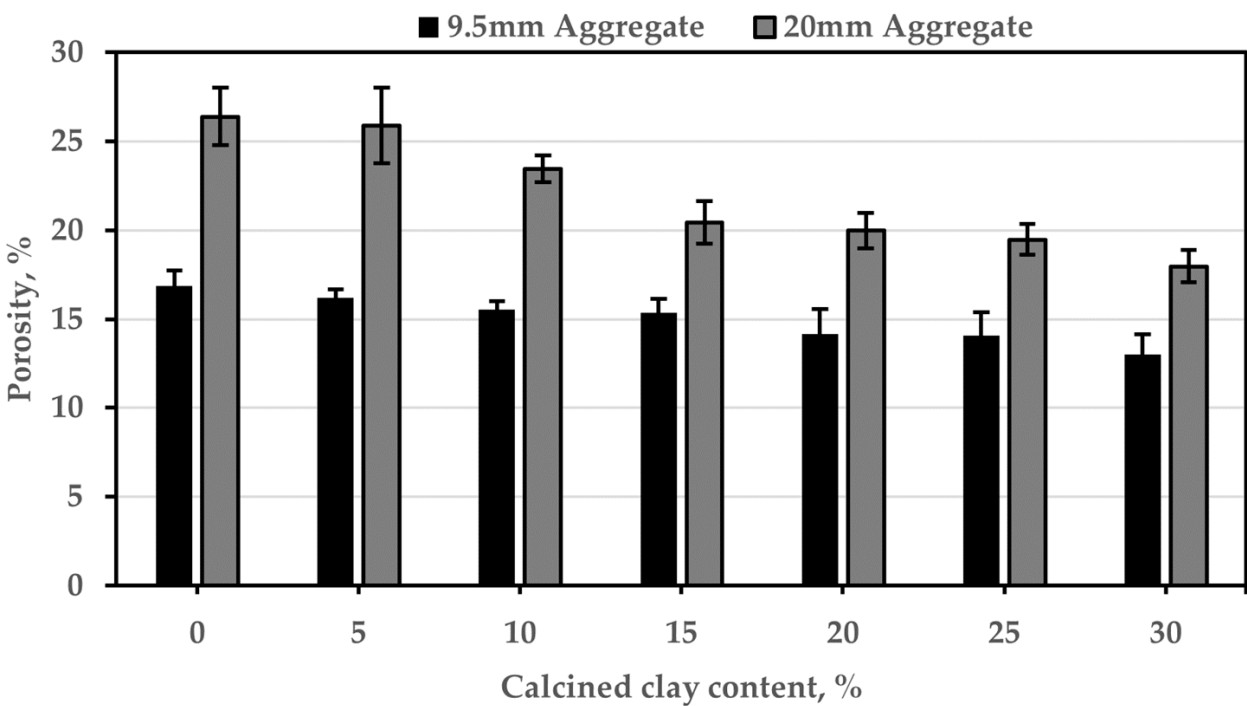

**Figure 6.** Porosity of pervious concrete.

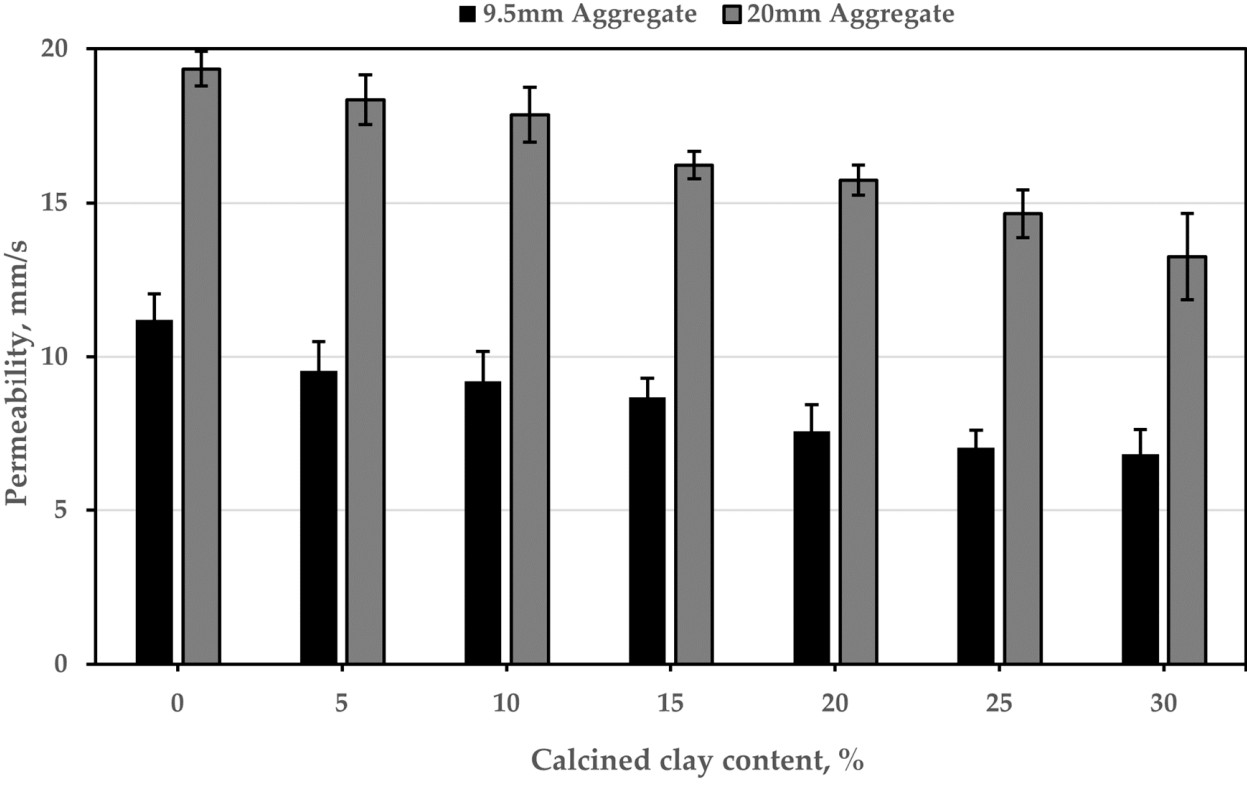

**Figure 7.** Permeability of pervious concrete.

### 3.3. Mechanical Properties

### 3.3.1. Compressive Strength

The compressive strength performance of calcined clay pervious concrete is shown in Figure 8. Compressive strength values obtained for 9.5 mm aggregate concrete were observed to be 7.7% greater than those of the 20 mm aggregate concrete. The larger aggregate sizes create pores between the paste and aggregates, leading to enhanced porosity and strength loss [38,40]. Again, larger aggregate sizes increase the fracture energy and fracture process zones, thereby resulting in expanded cracks and strength loss [41,42]. Compressive strength values, for both aggregate sizes, steadily improved with the addition of 5–20% calcined clay. With the incorporation of 20% calcined clay, the compressive strength improved by 12.7% and 16%, respectively, as compared to the 2 reference concrete mixes. The strength increase was due to the endothermic reaction between the constituents of the calcined clay and the cement to further generate cementitious compounds such as calcium silicate hydrates needed for strength formation during cement hydration [43–45]. Beyond 20% calcined clay inclusion, the compressive strength declined for both mixes, consistent with the results reported by Singh and Murugan [12]. This reduction in compressive strength is due to the clinker dilution effect [46]. A reduction in cement content impacts the pozzolanic reaction, leading to a shortage of the formation of C-S-H [47,48]. Therefore, raising the calcined clay content beyond 20% overburdens the concrete with excess unreacted $SiO_2$ and $Al_2O_3$, which do not contribute to strength development [49,50].

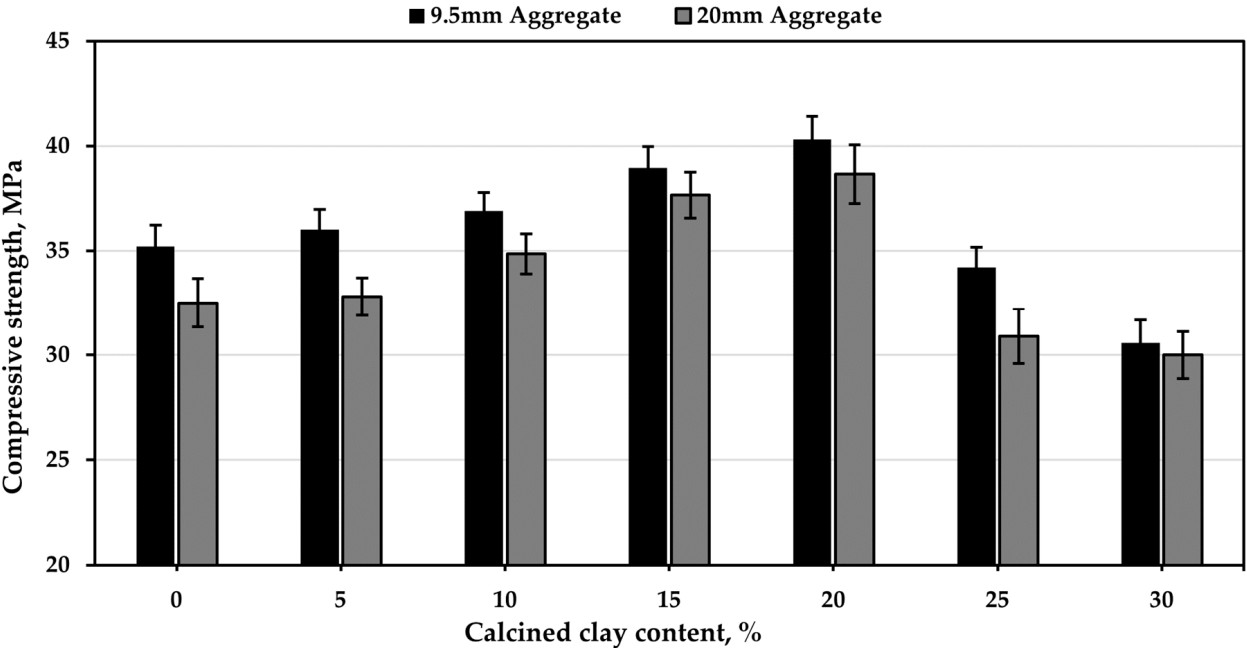

**Figure 8.** Compressive strength of pervious concrete.

### 3.3.2. Splitting Tensile Strength

The trend of results obtained for split tensile strength was consistent with that of compressive strength. The tensile strength increased as the aggregate size increased from 9.5 mm to 20 mm. Calcined clay significantly impacted the splitting tensile strength irrespective of the aggregate size. For pervious concrete prepared with 9.5 mm aggregate, the tensile strength increased by 8.4%, 15.7%, 26.6%, and 35.4% when cement was replaced with 5%, 10%, 15%, and 20% calcined clay, respectively. Percentage increments for 20 mm aggregate concrete were 8.9%, 10.5%, 24.7%, and 35.7%, respectively. An optimum splitting tensile strength was found at 20% replacement, which recorded 35.5% and 35.7% increases, respectively. Results for splitting tensile strength are shown in Figure 9.

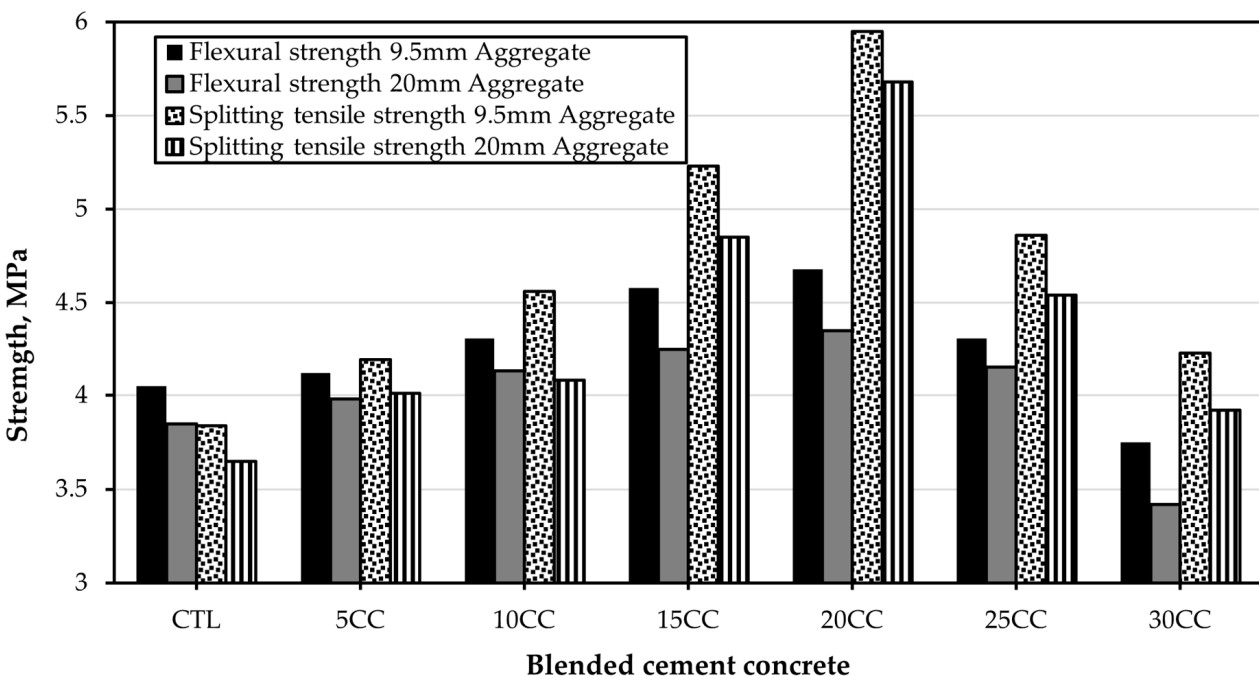

**Figure 9.** Tensile and flexural strength of pervious concrete.

### 3.3.3. Flexural Strength

Flexural strength, also a measure of tensile strength, is the ability of the concrete beam or slab to overcome bending moments [51]. The results of the flexural strength test, conducted after 28 days of water curing, are shown in Figure 9. It was observed that the variables (i.e., aggregate sizes resulting in porosity) that influenced compressive strength also affected the flexural strength development of the pervious concrete [52,53]. The flexural strength was found to be greater in 9.5 mm aggregate concrete than in 20 mm aggregate concrete. Furthermore, the incorporation of the pervious concrete with 5–20% of calcined clay increased the flexural strength by 1.7%, 6.0%, 11.6%, and 13.5% for 9.5 mm aggregate concrete and 3.3%, 6.8%, 9.2%, and 11.5%, respectively, for 20 mm aggregate pervious concrete. Beyond 20% calcined clay replacement, all concrete samples suffered a decline in flexural strength. The effect of calcined clay on flexural strength is shown in Figure 9.

### 3.4. Durability Properties

### 3.4.1. Ultrasonic Pulse Velocity (UPV)

Evaluation of the differences in the mechanical strength of concrete can be conducted by measuring the ultrasonic pulse velocity [54]. UPV technology is also used to detect voids as well as other discontinuities in concrete [55]. The UPV of concrete is significantly affected by the elasticity and mechanical properties of the concrete [56]. The influence of calcined clay and aggregate size on ultrasonic pulse velocity is illustrated in Figure 10. Samples modified with calcined clay of 5–20% experienced increasing UPV with increasing percentage replacement. The optimum UPV data were recorded at 20% calcined clay replacement, about 0.56% (for the 9.5 mm aggregate concrete) and 0.8% (for the 20 mm aggregate concrete) more than those of the reference concrete samples. Generally, the 20 mm aggregate concrete specimens recorded lower UPV values than the 9.5 mm aggregate concrete specimens. Higher UPV values of the calcined clay pervious concrete are an indication of the lower porosity and continuity of the samples, which may not be ideal for the purposes of pervious concrete. Similar results were reported by Dabbaghi et al. [57] and Nagrockiene et al. [58].

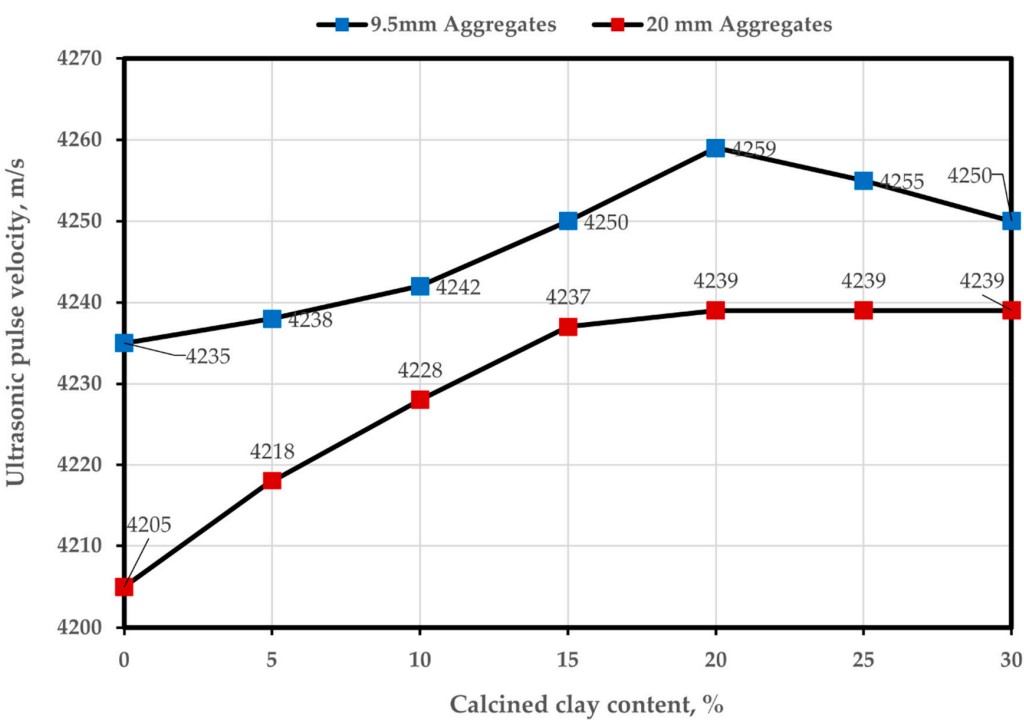

**Figure 10.** Ultrasonic pulse velocity measurements.

### 3.4.2. Thermal Conductivity

Figure 11 shows the thermal conductivity measurements of pervious concrete containing calcined clay. Thermal conductivity reductions recorded between 5 and 30% calcined clay replacement were 4%, 6.3%, 26.3%, 30.2%, 37.5%, and 51.2. Similarly, the thermal conductivity of samples containing 20 mm aggregates decreased with increasing calcined clay content, obtaining a similar range of values as the 9.5 mm aggregates. Clearly, aggregate size had little influence on thermal conductivity.

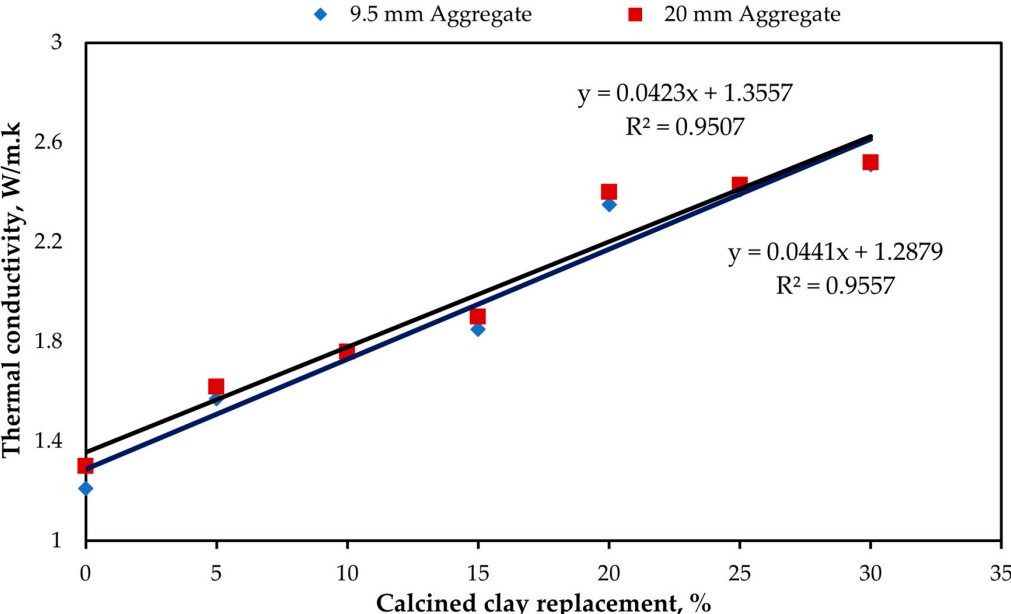

**Figure 11.** Thermal conductivity measurements.

The decrease in thermal conductivity was due to the pozzolanic reaction between calcined clay and portlandite from the cement to form a dense structure of calcium silicate hydrates, which serves as a thermal insulator [59].

### 3.4.3. Rapid Chloride Permeability Test (RCPT)

Figure 12 presents the findings for chloride permeability of the pervious concrete at 28 days. It was determined in the form of electric charge (in coulombs) travelling through the concrete samples. The charge passing through all samples was observed to decrease with calcined clay replacement. The results demonstrated that, compared to the reference cement, the incorporation of the calcined clay greatly improved the chloride penetration resistance. After 28 days, the amount of electric charge passing through the reference cement decreased by 18.5%, 19.1%, 20.4%, 23.6%, 26.4%, and 33.1% when 5%, 10%, 15%, 20%, 25%, and 30%, respectively, of the CEM I cement were replaced with the pozzolan. From Figure 12 an $R^2$ value of 0.7821 showed an imperfect relationship between calcined clay content and chloride permeability. Even though chloride permeability appeared to consistently decrease with increasing calcined clay content, other factors were likely to contribute to this trend of results. Again, 9.5 mm aggregate concrete outperformed the 20 mm aggregate concrete in terms of chloride resistance. This improvement in chloride resistance could be attributed to the improved density and less porosity in blended cement pervious concretes, as well as the reaction between the pozzolan and calcium hydroxide from the cement and the low electrical conductivity of blended cement concrete [6]. This trend is consistent with results obtained by Ramezanianpour and Jovein [60]. The scatter plots were plotted for identifying the relation of permeability and chloride penetration. It can be seen from Figure 13 that with the increase in permeability, chloride penetration increased. However, the goodness-of-fit between chloride penetration and permeability was lower, which can be mainly due to several other properties of pore structure other than permeability affecting chloride penetration directly.

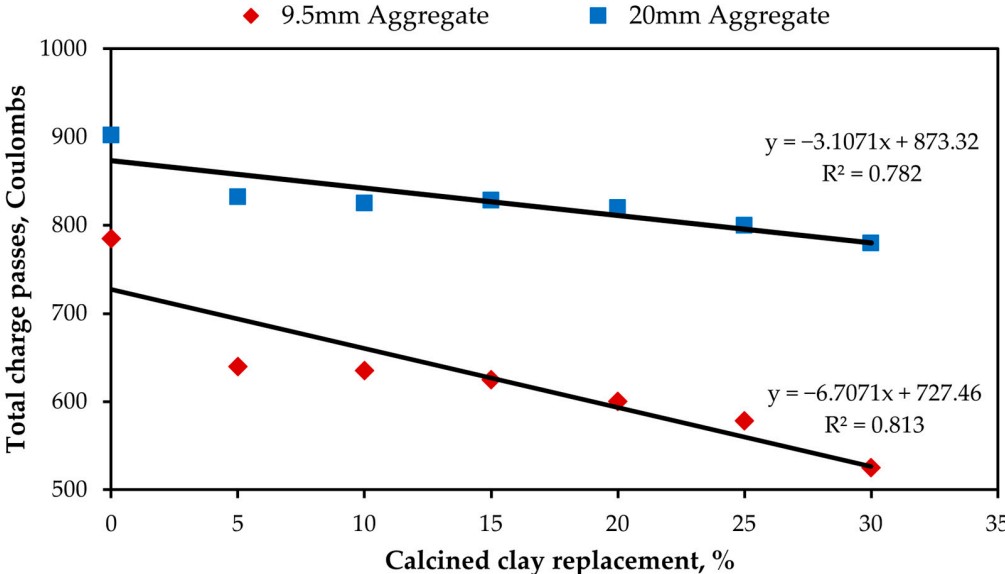

**Figure 12.** RCPT of pervious concrete.

### 3.4.4. Sulphate Resistance

The effect of 5%-$Na_2SO_4$ on blended cement pervious concrete, in terms of strength loss and weight loss, is presented in Figure 14. In addition, 5%-$Na_2SO_4$ was observed to have an adverse effect on the weight and, consequently, compressive strength of the pervious concrete. The reference sample recorded the highest weight and strength losses as compared to the blended cement samples, for all aggregate sizes. As the calcined clay content increased, weight loss and compressive strength loss increased. As seen in Figure 14, losses in compressive strength increased with increasing weight loss. $R^2$ values of 0.9558 and 0.9472 were obtained for 9.5 mm and 20 mm aggregate concretes, indicating a good fit. However, generally, concrete prepared with 20 mm aggregates suffered more weight and strength losses than the 9.5 mm concrete did. This could be linked to the

porosity of the two samples [61]. This trend is in line with results reported by other researchers [62,63].

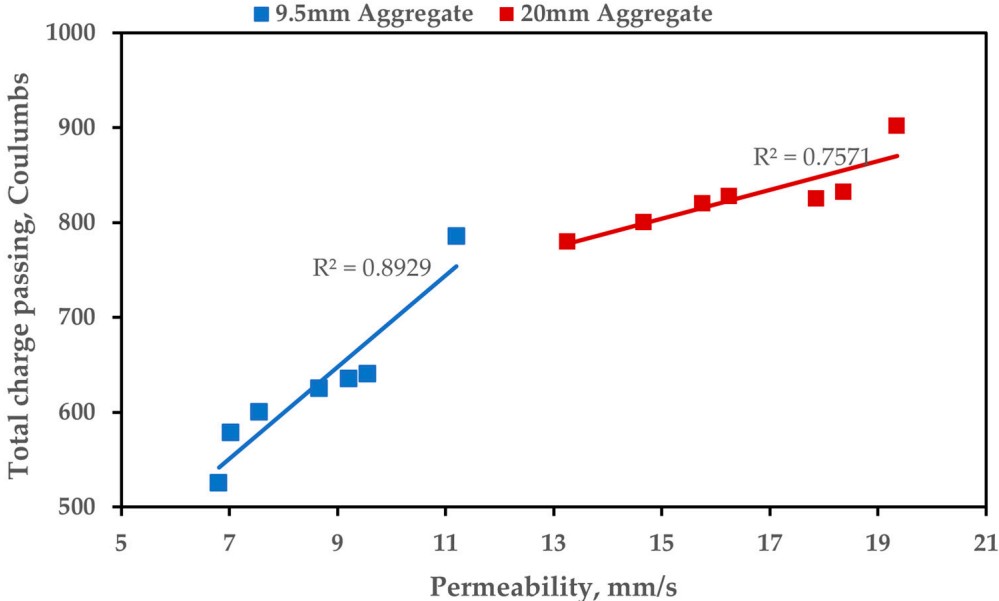

**Figure 13.** Relationship between RCPT and permeability.

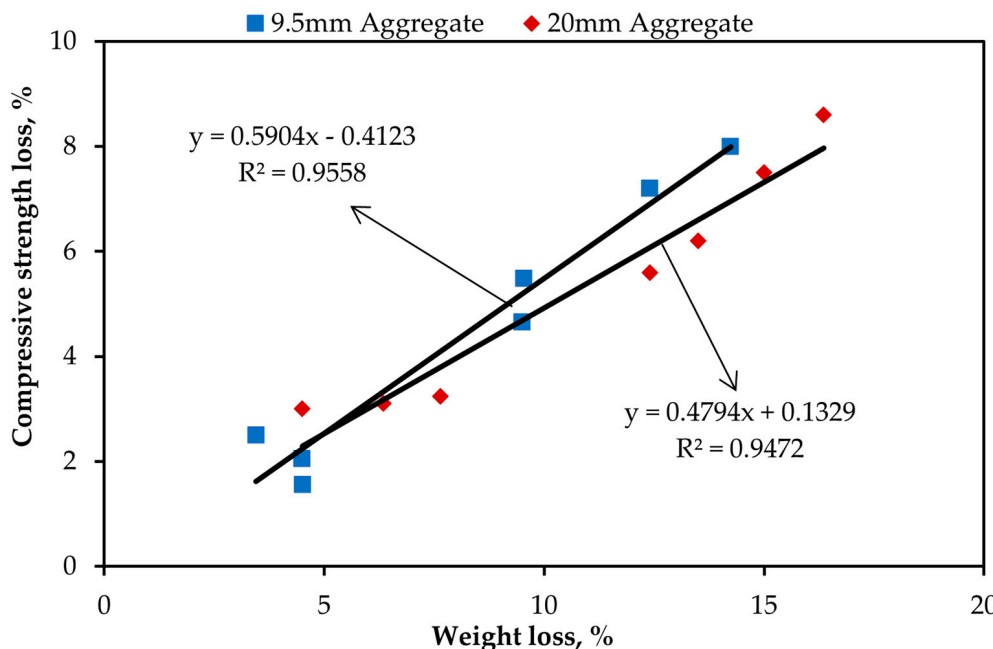

**Figure 14.** Relation between compressive strength loss and weight loss.

## 4. Conclusions

In this study, the behaviour of pervious concrete containing calcined clay pozzolan and two different aggregate sizes was investigated. CEM-I cement was replaced with calcined clay in weight percentages of 5%, 10%, 15%, 20%, 25%, and 30%. The influence of calcined clay on the mechanical and durability properties of pervious concrete was investigated and the following conclusions presented:

The density of the reference concrete was found to be lower than that of the calcined clay pervious concrete, which could be attributed to the fine particles of calcined clay occupying some of the voids in the concrete. Fewer voids often lead to denser concrete. Density was found to increase with increasing calcined clay concrete. This explains why, generally,

porosity and permeability decreased with increasing calcined clay content, irrespective of aggregate degradation. Comparatively, concrete prepared with 9.5 mm aggregates was observed to be denser and consequently exhibited less porosity and permeability than the 20 mm aggregate concrete did. Larger aggregate sizes took up a little more paste to coat its surfaces, leaving voids behind and reducing its density.

The mechanical properties of the pervious concrete displayed a similar trend. Compressive, flexural, and splitting tensile strengths increased as the calcined clay content increased up to 20%. Beyond 20% replacement, the tested strength declined. At 20% replacement, the compressive strength increased by 12.7% and 16% for 9.5 mm and 20 mm aggregates, respectively. The flexural strength improved by 13.5% and 11.5%, whereas the splitting tensile strength increased by 35.4% and 35.7%, respectively.

Generally, the 20 mm aggregate concrete specimens recorded lower UPV values than the 9.5 mm aggregate concrete specimens did. Samples modified with calcined clay of 5–20% experienced increasing UPV with increasing percentage replacement. In the RCPT test, the charges passing through the pervious concrete decreased as calcined clay increased. The charge flow was more pronounced in 20 mm aggregate concrete. The calcined clay concrete samples exhibited poorer thermal conductivity than the reference cement pervious concrete. The thermal conductivity of samples containing 20 mm aggregates decreased with increasing calcined clay content, obtaining a similar range of values as 9.5 mm aggregates. Clearly, aggregate size had little influence on thermal conductivity. Samples subjected to 5%-$Na_2SO_4$ for 90 days caused significant weight and compressive strength losses. However, losses in the reference cement were greater than those of the calcined clay pervious concrete. A linear relation indicated that compressive strength loss increased with increasing weight loss.

The optimum calcined clay replacement was found to be 20% by weight. Between the two aggregate sizes, 20 mm is recommended as porosity and permeability are crucial in designing pervious concrete, even though its mechanical properties were found to be slightly lower than those of 9.5 mm aggregates.

**Author Contributions:** Conceptualization, M.K.; methodology, M.K. and K.B.; software, M.K. and K.B.; validation, M.K.; formal analysis, K.B. and M.K.; investigation, K.B. and M.K.; resources, K.B. and M.K.; data curation, K.B. and M.K.; writing—original draft preparation, K.B.; writing—review and editing, M.K.; visualization, M.K.; supervision, M.K.; project administration, M.K.; funding acquisition, K.B. and M.K. All authors have read and agreed to the published version of this manuscript.

**Funding:** This research was self-funded by the authors and the APC was funded by MDPI open access publishing in Basel/Switzerland.

**Conflicts of Interest:** The authors declare no conflict of interest.

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
