# Peer review of "Influence of Calcined Clay Pozzolan and Aggregate Size on the Mechanical and Durability Properties of Pervious Concrete"

_jcs, doi:10.3390/jcs7050182_

Round 1
Reviewer 1 Report
This article investigated the Mechanical and Durability Properties of Pervious Concrete: The Role of Calcined Clay Pozzolan and Aggregate Size. Furthermore, the porosity, density, permeability, compressive strength, flexural strength, tensile strength, ultrasound pulse velocity, thermal conductivity, chloride penetration and sulphate resistance were examined. This article requires some improvements before it can be accepted for publication.
1. It is recommended to start the abstract with current issues in this research area and the purpose of the study.
2. Lines 105-111 are the repetition of the abstract. I suggest adding the novelty of this research.
3. Line 135: what is the scientific reason for selecting the two different sizes of aggregates? Explain the reason for this selection.
4. Why the calcium clay is limited to 30%?
5. Increasing calcium clay increased the workability? Why was the superplasticizer constant for all the mixtures?
6. Figure 12; what is the significance of linear equations presented in the figure?. Since the number of data points does not meet the minimum data points required for statistical analysis.
7. Please use the uniformity of splitting tensile strength. In some other locations, it has mentioned split tensile strength
8. Line 344: Results of the flexural strength test, conducted after 28 days of water curing, are shown in Figure 14. Figure 14 represents the splitting tensile strength. Please correct it
9. Figure 18: The R square value of 0.7821 is very low and how was the accuracy of the relationship between calcium clay replacement and total charge passes ensured?
Author Response
The response to reviewers' comments was attached

Reviewer 2 Report
Check attachment.

Author Response
The response the reviewers' comment was attchaed.

Round 2
Reviewer 1 Report
All comments are addressed adequately.
Reviewer 2 Report
The authors made the indicated recommendations.